# GVPO: Group Variance Policy Optimization for Large Language Model Post-Training

**Kaichen Zhang**[1,2]    **Yuzhong Hong**[2]    **Junwei Bao**[2,*]    **Hongfei Jiang**[2]
**Yang Song**[2]    **Dingqian Hong**[2]    **Hui Xiong**[1,3,*]

[1]Thrust of Artificial Intelligence, Hong Kong University of Science and Technology (Guangzhou)
[2]Zuoyebang Education Technology    [3]Department of Computer Science and Engineering, HKUST
kzhangbi@connect.ust.hk    eugene.h.git@gmail.com    baojunwei001@gmail.com
{jianghongfei,songyang,hongdingqian}@zuoyebang.com    xionghui@ust.hk

## Abstract

Post-training plays a crucial role in refining and aligning large language models to meet specific tasks and human preferences. While recent advancements in post-training techniques, such as Group Relative Policy Optimization (GRPO), leverage increased sampling with relative reward scoring to achieve superior performance, these methods often suffer from training instability that limits their practical adoption. As a next step, we present **Group Variance Policy Optimization (GVPO)**. GVPO incorporates the analytical solution to KL-constrained reward maximization directly into its gradient weights, ensuring alignment with the optimal policy. The method provides intuitive physical interpretations: its gradient mirrors the mean squared error between the central distance of implicit rewards and that of actual rewards. GVPO offers two key advantages: (1) it guarantees a **unique optimal solution**, exactly the KL-constrained reward maximization objective, (2) it supports flexible sampling distributions that **avoids importance sampling and on-policy limitations**. By unifying theoretical guarantees with practical adaptability, GVPO establishes a new paradigm for reliable and versatile LLM post-training.

## 1 Introduction

Large language models (LLMs) [59, 31], trained on extensive datasets, exhibit impressive general-purpose capabilities, yet their practical utility and alignment with human values depend critically on post-training [46] refinement. While pre-training [60, 33] equips LLMs with broad linguistic patterns, post-training techniques—such as supervised fine-tuning (SFT) [32] and reinforcement learning [30] from human feedback (RLHF) [3] are indispensable for adapting these models to specialized applications and ensuring their outputs align with ethical, safety, and user-centric standards.

> "The biggest lesson that can be read from 70 years of AI research is that general methods that leverage computation are ultimately the most effective, and by a large margin."
> — Rich Sutton, 2024 Turing Award winner

This principle outlined in *The Bitter Lesson* [43]—which advocates for scalable, computation-driven approaches—is exemplified by recent advances in post-training, particularly Group Relative Policy Optimization (GRPO) [39]. Diverging from conventional reinforcement learning frameworks [37] that depend on training a separate value function, GRPO directly optimizes advantage by standardizing reward scores across samples. This approach eliminates the need for an auxiliary value model, which typically demands computational resources comparable to those of the policy model itself. As a result, GRPO significantly reduces memory and computational overhead, enabling more efficient sampling and scalable training. Deepseek-R1 [11] leverages GRPO and achieves significant performance.

---

*Corresponding Authors. Code available at https://github.com/jszkc/GVPO

39th Conference on Neural Information Processing Systems (NeurIPS 2025).

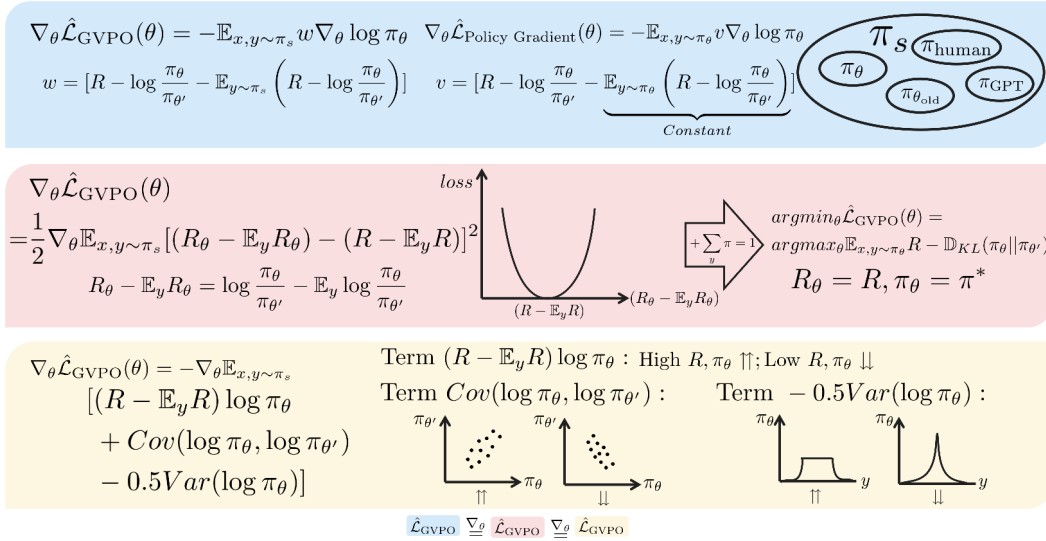

Figure 1: **Three equivalent loss functions of GVPO offer distinct interpretations**: (1) The Negative log-Likelihood perspective (top) illustrates that GVPO accommodates broader sampling distributions compared to conventional policy gradient methods; (2) The Mean Squared Error interpretation (middle) reveals GVPO's unique optimal solution, which simultaneously maximizes reward under a KL constraint; and (3) The Reinforcement Learning viewpoint (bottom) highlights GVPO's implicit regularization terms that ensure stable policy optimization. We assume $\beta = 1$ for simplicity.

However, GRPO has been documented to experience issues with training instability in prior literature [54, 28]. Specifically, GRPO is highly sensitive to its hyperparameters, such as the clip threshold and the KL-divergence coefficient. These limitations undermine the robustness of GRPO and hinder its broader practical adoption. As a next step, we propose **Group Variance Policy Optimization (GVPO)**, a novel approach for reliable and versatile LLM post-training.

Our analysis begins with a key observation: post-training algorithms—including but not limited to SFT, Reject Sampling [48], and GRPO—share a unified mathematical structure in their loss gradients [39, 10]. Specifically, each method's gradient can be expressed as a weighted sum of the gradients of the log-likelihoods of responses. This unified framework reveals that we can directly design weights to encode preferences—positive weights amplify gradients for favored responses, while negative weights suppress disfavored ones, with magnitudes modulating the strength of preference.

Motivated by the success of Direct Preference Optimization (DPO) [34]—which utilizes a closed-form link between reward models and the optimal policy under KL-divergence constraints [23]—we explore how to leverage this analytical relationship. A central obstacle arises from the partition function in the closed-form formula, which requires intractable expectation calculations over all possible responses. To address this, we identify a critical condition: *when the sum of assigned response weights within a prompt group equals zero, the partition function becomes invariant across compared responses*, effectively canceling out in the policy update rule. This insight eliminates the need for explicit estimation of the partition function, thereby enabling deployment of the closed-form optimal policy while retaining its theoretical advantages.

Based on the previous findings, we design GVPO's weighting scheme where the gradient weight of a response in a group is the difference between the central distance of implicit rewards-which derive from the current policy and the reference policy-and that of actual rewards, illustrated in Figure 1 (top panel). The loss is computable because the sum of weights in a prompt group equals zero.

We demonstrate that GVPO loss function carries physically meaningful interpretations. Specifically, we establish that its gradient equals that of a mean squared error loss measuring the discrepancy between implicit and actual reward central distances, illustrated in Figure 1 (middle panel).

Furthermore, the loss function in GVPO can be decomposed into three distinct components, as visualized in Figure 1 (bottom panel): (1) a group-relative reward term, (2) the variance of the current policy, and (3) the covariance between the current policy and a reference policy. The first component directly promotes advantage maximization by prioritizing responses with higher expected returns.

The covariance term acts as a regularizer, mitigating excessive deviations from the reference policy to ensure stable policy updates. Meanwhile, the variance term encourages moderate entropy, thereby naturally balancing exploration and exploitation. We systematically analyze GVPO's structural similarities with conventional policy gradient reinforcement learning methods.

We demonstrate that GVPO offers two key advantages:

- GVPO has a unique optimal solution, which coincides precisely with the optimal solution of the KL-constrained reward maximization. This guarantee confers a significant theoretical advantage over DPO. Prior work [4, 16] highlights that DPO may fail to converge to the optimal policy for the KL-constrained reward maximization problem, because of the inherent flaw of Bradley-Terry model [52]. In contrast, GVPO guarantees that its loss function is aligned with the original constrained optimization problem, ensuring convergence to the globally optimal policy. This theoretical robustness positions GVPO as a more reliable method for policy optimization.

- GVPO supports flexible sampling distributions that avoids importance sampling and on-policy limitations. Beyond the common practice of sampling from the previous step's policy, GVPO retains theoretical guarantees for the unique optimal solution under any sampling distribution satisfying a mild condition. This property provides a notable theoretical advantage over policy gradient methods [44]. Unlike on-policy approaches [41, 50], which require fresh trajectories for updates, GVPO facilitates off-policy training using reusable or heterogeneous datasets. Furthermore, in contrast to off-policy methods [39, 37] reliant on importance sampling, GVPO inherently avoids gradient explosion risks without introducing bias through clipping techniques.

As a result, GVPO emerges as a competitive online RL algorithm, capable of leveraging diverse data sources, sustaining stable policy updates, and preserving convergence to optimality.

## 2 Preliminary

Large language models take a prompt $x$ as input and generate a response $y$ as output. A policy $\pi_\theta(y_t|x, y_{<t})$ with parameter $\theta$ maps a sequence of tokens generated ($x$ and $y_{<t}$) to a probability distribution over the next token $y_t$. We also denote $\pi_\theta(y|x)$ as the probability of generating the response $y$ from $x$. A reward model $R(x, y)$ scores the response $y$ as the reply to the prompt $x$.

A reward model can explicitly be evaluation ratings of human beings; or a trainable function that implicitly reflects human preferences; or a predefined rule, such as correctness, accuracy.

The general purpose of post-training of large language model is summarized as following: Given an initial policy $\pi_{\theta_{init}}$, a dataset of prompts $x \sim D$, a reward model $R$, the objective is to train a new policy $\pi_\theta$ that generates responses with higher rewards, that is, maximize $E_{x \sim D, y \sim \pi_\theta(\cdot|x)} R(x, y)$.

### 2.1 Towards better computation leverage in post-training

The initial stage of large language model post-training typically involves Supervised Fine-Tuning (SFT) [32]. In this phase, a dataset comprising input prompts $x$ paired with exemplary responses $y$ is used to optimize the pre-trained model. The training minimizes the negative log-likelihood loss:

$$\mathcal{L}_{\text{SFT}}(\theta) = - \sum_{(x,y) \in \mathcal{D}} \log \pi_\theta(y|x) \tag{1}$$

Recent advancements, such as GRPO [39, 56], better leverage the computation by incorporating multiple sampled responses with their standardized reward scores as weights:

$$\mathcal{L}_{GRPO}(\theta) = - \sum_{(x,\{y_i\}) \in \mathcal{D}} \sum_{i=1}^{k} Clip(\frac{\pi_\theta(y_i|x)}{\pi_{\text{old}}(y_i|x)}) \frac{R_i - \overline{R}}{\sigma(R)} \log \pi_\theta(y_i|x) - \beta KL(\pi_\theta || \pi_{\text{ref}}) \tag{2}$$

Rewards below average lead to negative weights, minimizing their likelihoods. However, minimizing likelihood can result in unstable training [1]. Additionally, off-policy training of GRPO, which involves importance sampling with the weight $\frac{\pi_\theta}{\pi_{\theta_{old}}}$, becomes unstable when $\pi_\theta$ significantly deviates from $\pi_{\theta_{\text{ref}}}$, potentially causing gradient explosions. To mitigate these issues, GRPO uses gradient clipping and a KL constraint between the updated and reference policies. Nevertheless, empirical results [54, 17] show that GRPO still exhibits training instability, which undermines its performance.

## 2.2 Optimal solution to the KL-constrained reward maximization

In human preference alignment scenario [32], the ideal reward model would directly reflect human evaluative judgments. However, obtaining explicit human ratings is often unavailable in practice. Instead, contemporary approaches typically leverage pairwise response preferences $(x, y_w, y_l)$, where $y_w$ denotes the preferred response and $y_l$ the dispreferred response to prompt $x$, to approximate human preferences through reward model training. The resulting reward model subsequently enables policy optimization through the following KL-regularized objective:

$$max_{\pi_\theta} \mathbb{E}_{x \sim \mathcal{D}, y \sim \pi_\theta(y|x)}[R(x, y)] - \beta \mathbb{D}_{KL}[\pi_\theta(y|x)||\pi_{\theta'}(y|x)] \tag{3}$$

where $\beta > 0$ controls the divergence penalty from policy $\pi_{\theta'}$. In preference alignment scenario, $\pi_{\theta'}$ is set to a reference policy $\pi_{ref}$.

Rather than employing separate reward modeling and policy optimization stages, DPO [34] derives a single-stage training paradigm by exploiting the analytical relationship between optimal policies and reward functions. The optimal solution to Equation 3 satisfies:

$$\pi^*(y|x) = \frac{1}{Z(x)} \pi_{\theta'}(y|x) e^{R(x,y)/\beta} \tag{4}$$

which implies the corresponding reward function:

$$R(x, y) = \beta \log \frac{\pi^*(y|x)}{\pi_{\theta'}(y|x)} + \beta \log Z(x) \tag{5}$$

where $Z(x) = \sum_y \pi_{\theta'}(y|x) e^{R(x,y)/\beta}$ represents the partition function. DPO circumvents explicit computation of $Z(x)$ by substituting the reward expression from Equation 5 into the Bradley-Terry loss [5], yielding the final objective:

$$\mathcal{L}_{\text{DPO}}(\theta) = - \sum_{(x, y_w, y_l) \in \mathcal{D}} \log \sigma(\beta \log \frac{\pi_\theta(y_w|x)}{\pi_{ref}(y_w|x)} - \beta \log \frac{\pi_\theta(y_l|x)}{\pi_{ref}(y_l|x)}) \tag{6}$$

The success of DPO has been proven to be both efficient and effective. We attribute its achievements to its direct incorporation of the optimal policy's closed-form solution into the training objective.

## 2.3 Unified framework of post-training

As far as we know, post-training algorithms share a unified framework [39, 10], in which their losses' gradients share a same format:

$$\nabla_\theta \mathcal{L}(\theta) = - \sum_{(x, y_1, y_2, .., y_k) \in \mathcal{D}} \sum_{i=1}^{k} w_i \nabla_\theta \log \pi_\theta(y_i|x) \tag{7}$$

SFT only has one response per prompt, and its $w_1 = 1$. GRPO's essential weights are the standard scores of its rewards in a prompt group. Though it is not obvious for DPO, its gradients also share the same format, in which $w_w = \sigma(\beta \log \frac{\pi_\theta(y_l|x)}{\pi_{ref}(y_l|x)} - \beta \log \frac{\pi_\theta(y_w|x)}{\pi_{ref}(y_w|x)})$ and $w_l = -w_w$. Such the unified framework of post-training holds, because of the chain rule of derivatives.

---

**Algorithm 1** Group Variance Policy Optimization

---

**Require:** initial policy $\pi_\theta$; prompt distribution $\mathcal{D}$; hyperparameter $\beta$
1: **for** step $= 1, \ldots, n$ **do**
2:      Sample a batch $\mathcal{D}_b$ from $\mathcal{D}$
3:      Update the old policy model $\pi_{\theta_{old}} \leftarrow \pi_\theta$
4:      Sample $k$ responses $\{y_i\}_{i=1}^k \sim \pi_s(\cdot|x)$ for each prompt $x \in \mathcal{D}_b$
5:      Compute rewards $\{R(x, y_i)\}_{i=1}^k$ for every sampled response $y_i$ and prompt $x$
6:      Iteratively update policy $\pi_\theta$ by minimizing the GVPO loss (Equation 8, setting $\pi_{\theta'} = \pi_{\theta_{old}}$)
7: **end for**
8: **Return** $\pi_\theta$

---

# 3 Group Variance Policy Optimization

## 3.1 Motivation

The unified post-training framework (Equation 7) indicates that response preferences can be directly incorporated through the assignment of weights $w_i$.

To determine appropriate weights, we draw inspiration from the success of DPO. In particular, we aim to exploit the closed-form relationship between rewards and the optimal solution to the KL-constrained reward maximization objective: $R_\theta(x,y) = \beta \log(\pi_\theta(y|x)/\pi_{\theta'}(y|x)) + \beta \log Z(x)$.

However, the closed-form formula contains a partition function $Z(x)$ that is expensive to estimate in practice, because the function requires calculating the expectation of all possible responses.

To address this issue, we identify a critical condition: when the sum of assigned response weights within a prompt group equals zero, $\sum_{i=1}^{k} w_i = 0$, the partition function becomes invariant across responses: $\sum_{i=1}^{k} w_i R_\theta(x, y_i) = \sum_{i=1}^{k} w_i \beta \log(\pi_\theta(y_i|x)/\pi_{\theta'}(y_i|x))$.

## 3.2 Method

Build on this insight, we propose Group Variance Policy Optimization (GVPO), whose gradient weight $w_i$ is the difference between the central distance of implicit rewards-which derive from policy $\pi_\theta$ and policy $\pi_{\theta'}$-and that of actual rewards. Formally, GVPO's gradient $\nabla_\theta \mathcal{L}_{\text{GVPO}}(\theta) =$

$$-\beta \sum_{(x,\{y_i\}) \in \mathcal{D}} \sum_{i=1}^{k} [(R(x, y_i) - \overline{R(x, \{y_i\})}) - \beta(\log \frac{\pi_\theta(y_i|x)}{\pi_{\theta'}(y_i|x)} - \overline{\log \frac{\pi_\theta(\{y_i\}|x)}{\pi_{\theta'}(\{y_i\}|x)}})] \nabla_\theta \log \pi_\theta(y_i|x)$$

(8)

where $\overline{R(x, \{y_i\})} = \frac{1}{k} \sum_{i=1}^{k} R(x, y_i)$, and $\overline{\log \frac{\pi_\theta(\{y_i\}|x)}{\pi_{\theta'}(\{y_i\}|x)}} = \frac{1}{k} \sum_{i=1}^{k} \log \frac{\pi_\theta(y_i|x)}{\pi_{\theta'}(y_i|x)}$. We note that GVPO's gradient satisfies $\sum_{i=1}^{k} w_i = 0$. Algorithm 1 shows our proposed algorithm.

We demonstrate that GVPO's objective carries physically meaningful interpretations:

$$\nabla_\theta \mathcal{L}_{\text{GVPO}}(\theta)$$

$$= - \sum_{x, \{y_i\}} \sum_{i=1}^{k} [(R(x, y_i) - \overline{R(x, \{y_i\})}) - (R_\theta(x, y_i) - \overline{R_\theta(x, \{y_i\})})] \nabla_\theta \beta \log \pi_\theta(y_i|x)$$

$$= - \sum_{x, \{y_i\}} \sum_{i=1}^{k} [(R(x, y_i) - \overline{R(x, \{y_i\})}) - (R_\theta(x, y_i) - \overline{R_\theta(x, \{y_i\})})] \nabla_\theta R_\theta(x, y_i)$$

$$= - \sum_{x, \{y_i\}} \sum_{i=1}^{k} [(R(x, y_i) - \overline{R(x, \{y_i\})}) - (R_\theta(x, y_i) - \overline{R_\theta(x, \{y_i\})})] \nabla_\theta (R_\theta(x, y_i) - \overline{R_\theta(x, \{y_i\})})$$

$$= \frac{1}{2} \nabla_\theta \sum_{x, \{y_i\}} \sum_{i=1}^{k} [(R_\theta(x, y_i) - \overline{R_\theta(x, \{y_i\})}) - (R(x, y_i) - \overline{R(x, \{y_i\})})]^2$$

The first and second steps hold because $\beta \log Z(x)$ can cancel out. The second step holds because $\sum_{i=1}^{k} w_i \nabla_\theta \overline{R(x, \{y_i\})} = 0$. The third step holds because $\nabla_x f(x)^2 = 2f(x) \nabla_x f(x)$.

Essentially, we have established that GVPO's gradient mathematically equals that of a mean squared error loss measuring the discrepancy between implicit and actual reward central distances. Intuitively, when implicit rewards equal actual rewards or with a constant group shift, the GVPO's loss is minimized. This interpretation also implies that the response with higher actual rewards in a group is also encouraged to have higher implicit rewards, indicating higher $\log \frac{\pi_\theta(y_i|x)}{\pi_{\theta'}(y_i|x)}$.

Furthermore, by rearranging the mean squared error loss, we can derive a variance-based formulation, which represents the "**Variance**" term in the name GVPO:

$$\frac{1}{2} \nabla_\theta \sum_{x, \{y_i\}} \sum_{i=1}^{k} [(R_\theta(x, y_i) - R(x, y_i)) - \overline{R_\theta(x, \{y_i\}) - R(x, \{y_i\})}]^2$$

### 3.3 Theoretical guarantee

We show that GVPO has an unique optimal solution, and this unique optimal solution is exactly the optimal solution of reward maximization with KL constraint (Equation 4). Formally,

**Theorem 3.1.** *The unique optimal policy that minimizes $\hat{\mathcal{L}}_{GVPO}(\theta)$, defined as*

$$\hat{\mathcal{L}}_{GVPO}(\theta) = \mathbb{E}_{x\sim\mathcal{D}}\mathbb{E}_{y\sim\pi_s(\cdot|x)}[(R_\theta(x,y) - \mathbb{E}_{y\sim\pi_s}R_\theta(x,y)) - (R(x,y) - \mathbb{E}_{y\sim\pi_s}R(x,y))]^2 \quad (9)$$

*, is given by $\pi_\theta(y|x) = \pi^*(y|x) = \frac{1}{Z(x)}\pi_{\theta'}(y|x)e^{R(x,y)/\beta}$ for $\pi_s = \pi_{\theta'}$.*

We prove the theorem by establishing both necessity and sufficiency, provided in Appendix B.1.

Theorem 3.1 implies that the parameters minimizing $\hat{\mathcal{L}}_{\text{GVPO}}(\theta)$—guaranteed to be the sole global optimum—also maximize the expected rewards while maintaining proximity to a reference policy.

The uniqueness of the solution ensures the optimization landscape is well-behaved, avoiding sub-optimal local minima and guaranteeing convergence to a single, interpretable policy that optimally balances reward maximization with behavioral consistency relative to the reference. Consequently, this theorem bridges GVPO's practical algorithmic performance with theoretical guarantees.

**Theorem 3.2.** *The Theorem 3.1 also holds for any sampling distribution $\pi_s$ satisfying $\forall x, \{y|\pi_{\theta'}(y|x) > 0\} \subseteq \{y|\pi_s(y|x) > 0\}$.*

Beyond the conventional practice of sampling from the reference policy ($\pi_s = \pi_{\theta'}$), GVPO retains the theoretical guarantee of a unique optimal solution under any sampling distribution that satisfies a mild condition. This condition is readily met by any policy $\pi$ where $\pi(y, x) > 0$, a criterion inherently fulfilled by contemporary LLM policies utilizing softmax decoding.

The Theorem 3.2 of GVPO opens a new methodological avenue for off-policy LLM post-training. Prior off-policy methods have relied heavily on importance sampling [47], which suffers from two key limitations: (1) when $\pi_\theta$ diverges substantially from $\pi_s$, the importance weight $\frac{\pi_\theta}{\pi_s}$ can become either excessively large or vanishingly small, destabilizing training; and (2) when sampling involves heuristic or non-parametric components, $\pi_s$ becomes intractable to compute, thereby prohibiting techniques such as experience replay [9] in modern LLM post-training. In contrast, GVPO supports highly flexible off-policy sampling strategies while maintaining strong theoretical guarantees, enabling more robust and practical post-training paradigms for large language models.

### 3.4 Discussions with DPO

We begin by analyzing the foundational commonality between GVPO and DPO: both methods integrate the closed-form solution to the reward maximization problem under a KL divergence constraint into their training objectives. This integration establishes a direct relationship between the learned policy $\pi_\theta$ and the implicit reward function $R_\theta$, yielding two key advantages:

- It ensures an optimization process that inherently respects the KL divergence constraint, thereby preventing excessive deviation of the policy $\pi_\theta$ from the reference policy $\pi_{ref}$.

- It reduces the joint optimization over policies and rewards to a simpler problem focused solely on rewards. The latter is more tractable, as it requires only aligning the implicit rewards $R_\theta(x, y)$ with the true reward function $R(x, y)$.

To design effective methods leveraging this closed-form solution, two critical insights emerge:

1. **Computational Tractability**: The method must avoid intractable terms such as the partition function $Z(x)$. For instance, a naive loss $\mathcal{L} = \sum(R_\theta(x, y) - R(x, y))^2$ fails because $R_\theta(x, y)$ implicitly depends on $Z(x)$, which is computationally infeasible to estimate. DPO circumvents this by adopting the Bradley-Terry preference model, where $Z(x)$ cancels out in pairwise comparisons. GVPO proposes a novel *zero-sum property* across groups of responses, enabling cancellation of $Z(x)$ in broader multi-sample scenarios.

2. **Alignment with Desired Optimality**: The loss function must enforce meaningful convergence. For example, minimizing $\mathcal{L} = \sum\left(\beta\log\frac{\pi_\theta(x,y)}{\pi_{ref}(x,y)} - R(x,y)\right)^2$ yields a suboptimal solution

$R_\theta(x, y) = R(x, y) + \beta \log Z(x)$, which deviates from the true reward $R(x, y)$. A well-designed objective must avoid such misalignment. The method should adapt to available supervision. DPO leverages pairwise preference data without explicit rewards, while GVPO generalizes to group-wise responses with reward signals.

Moreover, GVPO demonstrates stronger theoretical robustness compared to DPO:

- Prior work [4, 16] highlights that DPO may fail to converge to the optimal policy for the KL-constrained reward maximization problem, because of the inherent flaw of Bradley-Terry model [52]. This arises because the DPO loss admits multiple minimizers, and its correlation with the true reward objective can diminish during training [45].

- In contrast, as formalized in Theorem 3.1 and Theorem 3.2, GVPO guarantees that its loss function is aligned with the original constrained optimization problem, ensuring convergence to the globally optimal policy. This theoretical robustness positions GVPO as a more reliable method for policy optimization in practice.

### 3.5 Discussions with GRPO and Policy Gradient methods

**Structural similarities**. Seeing the forest for the tree, we compare GVPO not only with GRPO but also with the broader family of policy gradient-based RL methods, beginning with their superficial structural similarities.

For simplicity, we assume $\beta = 1$ without loss of generality. Then $\hat{\mathcal{L}}_{\text{GVPO}}(\theta) \overset{\nabla_\theta}{=}$

$$\mathbb{E}_{x \sim \mathcal{D}, y \sim \pi_s(\cdot|x)}[(R_\theta(x, y) - \mathbb{E}_y R_\theta(x, y))^2 - 2(R(x, y) - \mathbb{E}_y R(x, y))R_\theta(x, y)]$$

$$\overset{\nabla_\theta}{=} \mathbb{E}_{x,y}[Var(\log \pi_\theta) - 2Cov(\log \pi_\theta, \log \pi_{\theta'}) - 2(R(x, y) - \mathbb{E}_y R(x, y)) \log \pi_\theta(y|x)] \tag{10}$$

$$= -2\mathbb{E}_{x,y}[(R(x, y) - \mathbb{E}_y R(x, y)) \log \pi_\theta(y|x) + Cov(\log \pi_\theta, \log \pi_{\theta'}) - 0.5 Var(\log \pi_\theta)]$$

where $Var(\log \pi_\theta) = (\log \pi_\theta(y|x) - \mathbb{E}_y \log \pi_\theta(y|x))^2$ and $Cov(\log \pi_\theta, \log \pi_{\theta'}) = (\log \pi_\theta(y|x) - \mathbb{E}_y \log \pi_\theta(y|x))(\log \pi_{\theta'}(y|x) - \mathbb{E}_y \log \pi_{\theta'}(y|x))$. As shown in Equation 10,

- the term $(R(x, y) - \mathbb{E}_y R(x, y)) \log \pi_\theta(y|x)$ encourages advantage maximization. Unlike conventional policy gradient methods that rely on explicit value function approximation [36], GRPO directly optimizes advantage by standardizing reward scores across samples. A distinction lies in GVPO's omission of standard deviation normalization.[2] Prior research [29] has also demonstrated that such scaling introduces bias by conflating prompt-level difficulty with reward signals.

- the term $Cov(\log \pi_\theta, \log \pi_{\theta'})$ serves to constrain deviations of the policy $\pi_\theta$ from policy $\pi_{\theta'}$, corresponding to $\mathbb{D}_{\text{KL}}[\pi_\theta||\pi_{\theta'}]$. Moreover, in GVPO, where $\pi_{\theta'} = \pi_{\theta_{\text{old}}}$, this term essentially aligns with the trust-region constraint [35], that ensures robustness between policy updates.

- the term $Var(\log \pi_\theta)$ strikes a balance between exploration and exploitation. We juxtapose this term with entropy regularization $-\mathbb{E}_y \log \pi(y|x)$ [2].

  - Increasing entropy encourages diversity by driving the policy toward a uniform distribution, but risks suppressing the likelihood of high-quality responses. Conversely, reducing entropy concentrates probability mass on a narrow set of outputs, diminishing diversity and potentially inducing entropy collapse. Consequently, entropy regularization proves highly sensitive to its coefficient, complicating practical implementation.

  - In contrast, $Var(\log \pi_\theta)$ circumvents this issue without requiring ad-hoc tuning by enabling scenarios where some responses receive zero probability, while other responses retain comparable probabilities. As the term $R - \overline{R}$ maximizes advantage, undesirable responses will receive zero probability, while favorable responses will maintain similar probabilities.

  - To illustrate the benefit, consider a toy example involving generation over the tokens $a, b, c, d, e$, where $\pi_\theta(a) = \pi_\theta(b) = 0$ and $\pi_\theta(c) = \pi_\theta(d) = \pi_\theta(e) = 1/3$. In this case, $Var(\log(\pi_\theta)) = 0$, indicating minimum variance and thus no penalty on this distribution. In contrast, entropy regularization only reaches its minimum either when the distribution is uniform or when it is one-hot, depending on whether one encourages higher or lower entropy.

---

[2]This omission is not a heuristic design but rather a consequence of the theoretical formulation.

**In-depth similarities**. In addition to structural similarities, we now highlight their key theoretical and practical distinctions. Modern policy gradient methods [35, 37, 39] optimize the expected reward under the current policy $\pi_\theta$ while constraining updates to avoid excessive deviation from the previous policy $\pi_{\theta_{\text{old}}}$. This is typically achieved by optimizing a objective that combines the reward $R(x, y)$ and a KL-divergence penalty term $\mathbb{D}_{\text{KL}}[\pi_\theta || \pi_{\theta_{\text{old}}}]$, yielding the gradient expression:

$$\nabla_\theta[\mathbb{E}_{x,y\sim\pi_\theta(y|x)}[R(x,y)] - \mathbb{D}_{\text{KL}}[\pi_\theta||\pi_{\theta_{\text{old}}}]] = \nabla_\theta\mathbb{E}_x\sum_y \pi_\theta(y|x)(R(x,y) - \log\frac{\pi_\theta(y|x)}{\pi_{\theta_{\text{old}}}(y|x)})$$

$$= \mathbb{E}_{x,y\sim\pi_\theta(y|x)}(R(x,y) - \log\frac{\pi_\theta(y|x)}{\pi_{\theta_{\text{old}}}(y|x)} - 1)\nabla_\theta\log\pi_\theta(y|x)$$

(11)

However, estimating this expectation requires on-policy sampling from $\pi_\theta(y|x)$, leading to low sample efficiency—a well-documented limitation of policy gradient methods. Reusing stale samples from prior policies introduces bias, degrading optimization stability and final performance.

To mitigate this, prior works [35, 37, 39] employ importance sampling, rewriting Equation 11 as:

$$\mathbb{E}_{x,y\sim\pi_{\theta_{\text{old}}}(y|x)}\frac{\pi_\theta(y|x)}{\pi_{\theta_{\text{old}}}(y|x)}\left(R(x,y) - \log\frac{\pi_\theta(y|x)}{\pi_{\theta_{\text{old}}}(y|x)} - 1\right)\nabla_\theta\log\pi_\theta(y|x).$$

(12)

This allows off-policy gradient estimation using samples from $\pi_{\theta_{\text{old}}}$. However, $\frac{\pi_\theta(y|x)}{\pi_{\theta_{\text{old}}}(y|x)}$ becomes unstable when $\pi_\theta$ deviates significantly from $\pi_{\theta_{\text{old}}}$, risking gradient explosion. Heuristics like gradient clipping [37] address this at the cost of biased gradient estimates, undermining theoretical guarantees.

GVPO circumvents these issues because it does not necessitate on-policy sampling in the first place. By rearranging Equation 11, we observe that the policy gradient can be expressed as:

$$\mathbb{E}_{x,y\sim\pi_\theta(y|x)}\left[R(x,y) - \log\frac{\pi_\theta(y|x)}{\pi_{\theta_{\text{old}}}(y|x)} - \mathbb{E}_{y\sim\pi_\theta(y|x)}\left(R(x,y) - \log\frac{\pi_\theta(y|x)}{\pi_{\theta_{\text{old}}}(y|x)}\right)\right]\nabla_\theta\log\pi_\theta(y|x),$$

where the baseline term (subtracted expectation) arises because $\mathbb{E}_{y\sim\pi_\theta}[c\nabla_\theta\log\pi_\theta(y|x)] = \nabla_\theta c = 0$ for any constant $c$. Crucially, the GVPO gradient generalizes this structure, $\nabla_\theta\hat{\mathcal{L}}_{\text{GVPO}}(\theta) =$

$$\mathbb{E}_{x,y\sim\pi_s(y|x)}\left[R(x,y) - \log\frac{\pi_\theta(y|x)}{\pi_{\theta_{\text{old}}}(y|x)} - \mathbb{E}_{y\sim\pi_s(y|x)}\left(R(x,y) - \log\frac{\pi_\theta(y|x)}{\pi_{\theta_{\text{old}}}(y|x)}\right)\right]\nabla_\theta\log\pi_\theta(y|x),$$

This reveals that classical policy gradient under trust-region constraint is a special case of GVPO gradient with $\pi_s = \pi_\theta$. As proven in Theorem 3.1 and Theorem 3.2, GVPO retains the same optimal solution as the policy gradient method while decoupling the sampling distribution $\pi_s$ from the learned policy $\pi_\theta$. GVPO's decoupling addresses two critical limitations:

1. **Sample Efficiency**: Unlike on-policy methods [41, 50], GVPO supports off-policy training with reusable or mixed data (e.g., expert demonstrations, historical policies, or model distillations).

2. **Stability**: By avoiding importance sampling weights $\frac{\pi_\theta}{\pi_{\theta_{\text{old}}}}$, GVPO eliminates gradient explosion risks without biased clipping.

By synergizing these advantages, GVPO emerges as a competitive online reinforcement learning algorithm capable of leveraging diverse data sources, sustaining stable policy updates, and preserving convergence to optimality—a combination previously unattained in prior policy gradient methods.

Table 1: Algorithm Performance Comparison on Mathematical Datasets

| Algorithm | AIME2024 | AMC | MATH500 | Minerva | Olympiadbench |
|---|---|---|---|---|---|
| Qwen2.5-Math-7B | 14.68 | 38.55 | 64.00 | 27.20 | 30.66 |
| +GRPO | 14.79 | 55.42 | 80.00 | 41.17 | 42.07 |
| +Dr.GRPO | 16.56 | 48.19 | 81.20 | 44.48 | 43.40 |
| +Remax | 17.19 | 60.24 | 82.00 | 40.44 | 45.19 |
| +Reinforce++ | 16.67 | 54.22 | 80.40 | 43.01 | 41.78 |
| **+GVPO** | **20.72** | **62.65** | **83.80** | **45.95** | **46.96** |

# 4 Experiments

**Task.** Following the established experimental setting of GRPO, we conduct a comprehensive evaluation on math reasoning. Specifically, we post-train the Qwen2.5-Math-7B model on Competition Math dataset [14] and assess performance on AIME2024 [26], AMC [26], Math500 [15], Minerva [25], and OlympiadBench [13]. For answer verification, we utilize the xVerify framework [6]. We adopt the $pass@1$ accuracy for all benchmarks except AIME2024, where we report $avg@32$ accuracy to account for its limited size (30 problems) and high difficulty.

In addition to math reasoning, we also evaluate GVPO on the summarization task in Appendix C.

**Setup.** To ensure a fair comparison across methods, we maintain identical experimental settings while only modifying the algorithmic component. For GVPO, we employ $\beta = 0.1$ and $\pi_s = \pi_{\theta_{\text{old}}}$ in the main experiment. For competing approaches, we utilize hyperparameters specified in their original publications. All experiments generate $k = 5$ responses per prompt. A comprehensive description of the training details is provided in Appendix A.1.

**Main Result.** Table 1 shows the main experiment result, which demonstrates that GVPO achieves the best performance, outperforming both the base model and other variants in all benchmarks [28, 27, 17], particularly in complex problem-solving scenarios. We attribute its effectiveness to its strong theoretical guarantees of convergence.

**Ablation on $\beta$.** Figure 2 analyzes the sensitivity of GVPO to variations in $\beta$. The results demonstrate little fluctuation in performance across $\beta$, suggesting GVPO exhibits robustness to this hyperparameter. This stability may reduce the need for exhaustive tuning and enhance its practical utility.

**Ablation on $k$.** Figure 3 examines how GVPO scales with $k$, evaluated on Qwen2.5-Math-1.5B. Top and bottom panels show results for MATH500 and AIME2024 respectively. GVPO consistently outperforms GRPO across all $k$ and demonstrates superior scalability. Notably, GVPO matches the AIME2024 performance of a 7B model on the 1.5B architecture through increased $k$, highlighting its potential for reducing inference costs in practice.

**Ablation on $\pi_s$.** Figure 4 investigates GVPO's versatility on sampling distributions, evaluated on Qwen2.5-Math-1.5B and MATH500. We propose a heuristic $\pi_s$ that mixes responses from $\pi_{\theta_{\text{old}}}$ with historical responses. Results demonstrate GVPO's robust performance across mixing proportions, highlighting: (1) this $\pi_s$ can reduce sampling costs during training, and (2) it suggests GVPO's potential to bridge modern LLM research with previous RL research on exploration strategies.

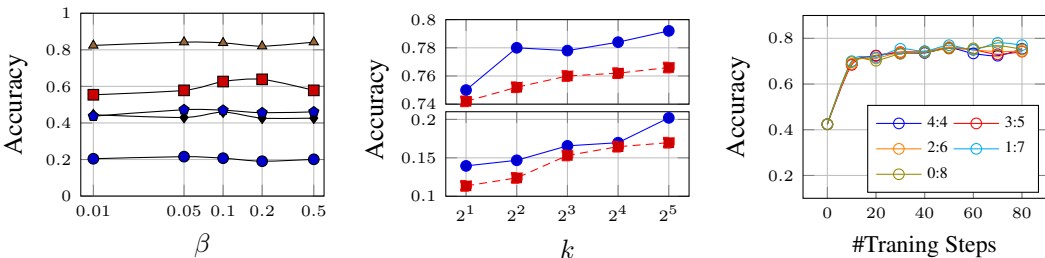

Figure 2: Ablation on $\beta$. Each line represents a dataset.

Figure 3: Ablation on $k$. Blue line: GVPO; Red line: GRPO.

Figure 4: Ablation on $\pi_s$. #(historical $y$) : #($y$ from $\pi_{\theta_{\text{old}}}$)

Table 2: Ablation on Regularization Terms

| Algorithm | AIME24 | AMC | MATH500 | Minerva | Olympiadbench |
|---|---|---|---|---|---|
| GVPO | 20.72 | 62.65 | 83.80 | 45.95 | 46.96 |
| GVPO - Var | 0.00 | 0.00 | 0.00 | 0.00 | 0.00 |
| GVPO - Cov | 0.00 | 0.00 | 0.00 | 0.00 | 0.00 |
| GVPO - Var - Cov | 7.19 | 39.76 | 73.00 | 34.93 | 35.70 |
| GVPO - Var + Entropy | 0.00 | 0.00 | 0.00 | 0.00 | 0.00 |
| GVPO - Var + Entropy (LR=1e-6) | 3.02 | 39.76 | 33.20 | 23.16 | 8.89 |

**Ablation on Regularization Terms.** In Section 3.5, we decompose the GVPO loss into three components: an advantage maximization term, a variance regularization term ($Var$) on the current policy, and a covariance regularization term ($Cov$) between the current and reference policies.

Table 2 presents the ablation results for these regularization terms. We first remove $Var(\log \pi_\theta)$ and $Cov(\log \pi_\theta, \log \pi_{\theta'})$ individually. In both cases, the model fails to converge and generates incoherent outputs, indicating that each term plays a crucial role in stabilizing training. When both regularization terms are removed simultaneously—reducing the objective to $R - \overline{R}$—the model initially converges but diverges after approximately 10% of the training steps, further confirming that these regularization components are essential to GVPO's stability.

**Study on $Var(\log \pi_\theta)$.** We further investigate the role of $Var(\log \pi_\theta)$ by replacing it with an entropy regularization term. As shown in Table 2, the model again fails to converge and produces incoherent outputs. Lowering the learning rate to $1e-6$ improves stability marginally, yet substituting it with entropy regularization still lead to divergence or suboptimal performance. These findings suggest that although entropy regularization serves a similar stabilizing purpose, it cannot fully substitute $Var(\log \pi_\theta)$. The degradation likely arises from entropy regularization being either too weak—insufficient to suppress extreme updates—or too strong—impeding convergence. This underscores a key limitation of entropy regularization: its sensitivity to coefficient tuning. In contrast, the coefficient for $Var(\log \pi_\theta)$ in GVPO is derived analytically via the optimal solution theorem, eliminating the need for manual tuning and enhancing overall robustness.

Table 3: Algorithm Performance Comparison with Different Random Seeds

| Algorithm | AIME2024 | AMC | MATH500 | Minerva | Olympiadbench |
|---|---|---|---|---|---|
| GRPO | 13.59±1.11 | 44.34±2.19 | 76.30±1.35 | 35.40±1.04 | 38.65±0.74 |
| **GVPO** | **15.56±1.05** | **45.78±2.05** | **77.36±0.98** | **36.03±1.15** | **39.58±1.08** |

**Robustness Check with Different Random Seeds.** We conducted additional experiments using 10 random seeds for both methods on Qwen2.5-Math-1.5B. The results show that GVPO consistently outperforms GRPO in overall performance while exhibiting comparable standard deviations, indicating stable results across different runs.

Table 4: Algorithm Performance Comparison with Llama-3.1-8B-Instruct

| Algorithm | AIME2024 | AMC | MATH500 | Minerva | Olympiadbench |
|---|---|---|---|---|---|
| Llama-3.1-8B-Instruct | 5.00 | 19.27 | 50.40 | 22.42 | 17.04 |
| +GRPO | 8.54 | 19.27 | 52.60 | 25.37 | 16.15 |
| **+GVPO** | **11.56** | **20.48** | **56.60** | **29.41** | **20.59** |

**Robustness Check with a Different Foundation Model.** To further assess generalization, we repeated the experiments using Llama-3.1-8B-Instruct as the foundation model. As shown in Table 5, GVPO again outperforms GRPO, reaffirming the robustness and effectiveness of our approach.

## 5 Related Work

This paper closely relates to the LLM post-training literature, including SFT [32, 8], RLHF [3], PPO [37], TRPO [35], DPO [34], GRPO [39], Dr.GRPO [28], Remax [27], Reinforce++ [17], DAPO [54]; and debates on whether RL improves LLM reasoning capacity [55], spurious reward issue [38], data contamination issue [51], and Pass@K optimization [56].

Beyond post-training, LLMs have demonstrated broad applicability across domains, including vision-language modeling [12], graph learning [24, 19], AI safety research [53], evaluation frameworks leveraging LLM-as-a-Judge paradigms [22, 21], retrieval-augmented generation [7], AI for Education [20, 49] and economics of AI [57, 58]. These broader developments underscore the promising role of post-training in enhancing the adaptability of LLMs across a growing range of applications.

## 6 Conclusion

In this paper, we present Group Variance Policy Optimization (GVPO). GVPO guarantees a unique optimal solution, exactly the KL-constrained reward maximization objective. Moreover, it supports flexible sampling distributions that avoids on-policy and importance sampling limitations. Through systematic comparisons with other prominent methods both theoretically and empirically, we establish GVPO as a new paradigm for reliable and versatile LLM post-training.

## Acknowledgments and Disclosure of Funding

We thank all the reviewers, the AC and program committee members for constructive feedback.

This work was supported in part by the National Key R&D Program of China (Grant No.2023YFF0725001), in part by the National Natural Science Foundation of China (Grant No.92370204), in part by the guangdong Basic and Applied Basic Research Foundation (Grant No.2023B1515120057), in part by the Education Bureau of Guangzhou.

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

## Limitation

~~In compliance with page limit requirement, this paper does not extensively discuss experimental results. Instead, we have prioritized an in-depth analysis of GVPO's essence, as we posit that such a discussion offers greater scholarly value to readers. We acknowledge this structural limitation and note that it will be addressed in subsequent revisions, particularly in the event of acceptance with an additional page allowance.~~


# Appendix

## A   Supplementary Experiment Information

### A.1   Experiment Settings

**Hyperparameter Recipe.** For each step, we sample $1024$ prompts from the training set and set the mini-batch size in each step to $256$. We repeat the whole training set for 10 epochs and set the warm-up ratio to $5\%$. We grid-search the learning rate in $\{5e-7,\ 1e-6, 5e-6, 1e-5\}$ and find $5e-6$ to be the best setting. We conduct the main experiment using an Deepseek-R1-like chat template on top of Qwen2.5-Math-7B as in [18]. In the ablation experiments, for faster training and GPU memory limitations, we use the original Qwen chat template on top of Qwen2.5-Math-1.5B.

**Compute Resources.** We conduct our experiments using a server with eight 80GB H800 GPU cards. For Qwen2.5-Math-7B experiments with $k = 5$, it takes 6 to 8 minutes per training step and approximately 12 hours per experiment. For Qwen2.5-Math-1.5B experiments with $k = 8$, it takes 4 to 5 minutes per training step and approximately 8 hours per experiment.

### A.2   Code Implementation

It is easy to implement GVPO based on open-source RL framework. For example[3], we show the minimum viable implementation of GVPO that only modifies a few line of GRPO loss in verl [40]:

```
def compute_policy_loss(old_log_prob, log_prob, advantages, eos_mask,
    **kwargs):
    scores = (log_prob * eos_mask).sum(dim=-1)
    scores_old = (old_log_prob * eos_mask).sum(dim=-1)
    advs = (advantages * eos_mask).sum(dim=-1) / eos_mask.sum(dim=-1)

    beta = 0.1
    k = scores.size(0)

    scores_new = scores.detach()
    loss = -1 * beta * scores * (advs - beta * ((scores_new -
        scores_new.mean()) - (scores_old - scores_old.mean())))

    return loss.sum()/(k-1)
```

Listing 1: A Simple GVPO Code Implementation

Moreover, we provide an implementation based on verl at https://github.com/jszkc/GVPO.

## B   Proofs

### B.1   Proof of Theorem 3.1

**Theorem 3.1**. *The unique optimal policy that minimizes $\hat{\mathcal{L}}_{GVPO}(\theta)$, defined as*

$$\hat{\mathcal{L}}_{GVPO}(\theta) = \mathbb{E}_{x\sim\mathcal{D}}\mathbb{E}_{y\sim\pi_s(\cdot|x)}[(R_\theta(x,y) - \mathbb{E}_{y\sim\pi_s}R_\theta(x,y)) - (R(x,y) - \mathbb{E}_{y\sim\pi_s}R(x,y))]^2$$

*, is given by $\pi_\theta(y|x) = \pi^*(y|x) = \frac{1}{Z(x)}\pi_{\theta'}(y|x)e^{R(x,y)/\beta}$ for $\pi_s = \pi_{\theta'}$.*

*Proof.* We prove the theorem by establishing both necessity and sufficiency.

**Necessity:** If $\pi_\theta(y|x) = \pi^*(y|x)$, then it is an optimal policy solution.

The loss function $\hat{\mathcal{L}}_{\text{GVPO}}(\theta)$ is non-negative because it represents the expectation of a squared term:

$$\hat{\mathcal{L}}_{\text{GVPO}}(\theta) = \mathbb{E}_{x,y}\left[\left((R_\theta(x,y) - \mathbb{E}_y R_\theta(x,y)) - (R(x,y) - \mathbb{E}_y R(x,y))\right)^2\right] \geq 0.$$

---

[3]Make sure that 1) each input batch correspond to the $k$ responses of a prompt and 2) the std normalizer in the GRPO advantage calculation has been removed.

When $\pi_\theta(y|x) = \pi^*(y|x)$, we have $R_\theta(x, y) = R(x, y)$. Substituting this into the loss function gives $\hat{\mathcal{L}}_{\text{GVPO}}(\theta) = 0$, confirming that $\pi^*$ achieves the minimum loss.

**Sufficiency:** If a policy $\pi_\theta$ is optimal, then $\pi_\theta(y|x) = \pi^*(y|x)$.

Assume for contradiction that there exists an optimal policy $\pi_\theta \neq \pi^*$. Since $\pi_\theta$ is optimal, $\hat{\mathcal{L}}_{\text{GVPO}}(\theta) = 0$. This implies:

$$(R_\theta(x, y) - \mathbb{E}_y R_\theta(x, y)) = (R(x, y) - \mathbb{E}_y R(x, y)), \quad \forall x, y \text{ s.t. } \pi_s(y|x) > 0$$

Rewriting $R_\theta$ and $R$ in terms of their respective policies:

$$\beta log \pi_\theta(y|x) - \mathbb{E}_y R_\theta(x, y) = \beta log \pi^*(y|x) - \mathbb{E}_y R(x, y).$$

Rearranging terms yields:

$$\pi_\theta(y|x) = \exp\left(\frac{\mathbb{E}_y[R_\theta(x, y) - R(x, y)]}{\beta}\right) \pi^*(y|x).$$

Since $\sum_{y \in \{y|\pi_{\theta'}(y|x) > 0\}} \pi_\theta(y|x) = \sum_{y \in \{y|\pi_{\theta'}(y|x) > 0\}} \pi^*(y|x) = 1$, we must have:

$$\sum_y \pi_\theta(y|x) = \exp\left(\frac{\mathbb{E}_y[R_\theta(x, y) - R(x, y)]}{\beta}\right) \sum_y \pi^*(y|x)$$

$$\implies \exp\left(\frac{\mathbb{E}_y[R_\theta(x, y) - R(x, y)]}{\beta}\right) = 1$$

Thus, $\pi_\theta(y|x) = \pi^*(y|x)$ for all $x, y$, contradicting the assumption $\pi_\theta \neq \pi^*$.

Since both necessity and sufficiency hold, the optimal policy is uniquely $\pi^*$. $\qquad\square$

## C  Alternative Task on Summarization

To evaluate GVPO in the context of alignment [4], we conduct experiments on a summarization task following the experimental setup of DPO [34]. Specifically, we used the Reddit dataset from [42] and the reward model OpenAssistant/reward-model-deberta-v3-large-v2. First, we fine-tuned Qwen2.5 1.5B using the dataset to obtain an SFT reference model. We then applied both DPO and GVPO for post-training. The training data was sampled from the reference model and scored using the reward model.

For evaluation, we considered:

- The average reward of generated summaries on the test split.
- Win rate against human-written demonstration answers, as scored by the reward model.
- Preference accuracy using human-labeled comparison data: both DPO and GVPO are trained via $R_\theta(x, y) = \beta \frac{\log \pi_\theta(y|x)}{\log \pi_{\text{ref}}(y|x)} + \beta \log Z(x)$. Given a preference pair $(y_w, y_l)$ where $y_w$ is preferred, we compute whether $\frac{\log \pi_\theta(y_w|x)}{\log \pi_{\text{ref}}(y_w|x)} > \frac{\log \pi_\theta(y_l|x)}{\log \pi_{\text{ref}}(y_l|x)}$.
- Benchmarking by a set of powerful LLMs.
- Human evaluations by crowd-sourcing workers. We hire three workers to label their preference on 100 prompts on Prolific platform.

Table 5: Algorithm Performance Comparison on Summarization

| Algorithm | Reward | Win Rate | Accuracy | Gpt-4o | Gemini-2.5-pro | Deepseek-R1 | Human |
|-----------|--------|----------|----------|--------|----------------|-------------|-------|
| SFT | 2.84 | 41.85% | — | — | — | — | — |
| +DPO | 4.83 | 68.28% | 60.43% | 31% | 40% | 25% | 34% |
| +GVPO | 5.75 | 79.49% | 64.93% | 69% | 60% | 75% | 66% |

The results indicate that GVPO outperforms DPO across these metrics, supporting the view that improvements in policy search methods are positively correlated with alignment quality.

---

[4] Thank Reviewer X8wt for sponsoring this study.

