# OpenReview forum: "GVPO: Group Variance Policy Optimization for Large Language Model Post-Training"
_NeurIPS.cc/2025/Conference — NeurIPS 2025 poster_

### Official Review · Reviewer_KXfz · 2025-06-15

**Clarity:** 2
**Significance:** 2
**Originality:** 2
**Rating:** 4
**Confidence:** 3

**Summary:**

This paper introduces a new loss function for post-training LLMs that incorporates KL-constrained reward maximization directly into its gradient updates. In particular, the paper first identifies that common post-training methods such as SFT, rejection sampling, GRPO, etc, have a unified structure, as a weighted sum of log-likelihood gradients. Then, improving on this structure, they propose a novel loss with weights as the difference between implicit rewards defined in terms of $\pi_{\theta}$ and $\pi_{\theta'}$ and actual rewards. The implicit reward expression is derived from the optimal solution to the KL-constrained reward maximization problem in line to the DPO framework. They then meaningfully reinterpret this as the GVPO loss which minizes the squared distance between implicit rewards and actual rewards.

They analyze the theoretical guarantees of this GVPO loss in terms of optimal solution, different sampling policy, convergence guarantees, and unbiasedness of the proposed estimator. They detail two discussion section where they this loss function with DPO and GRPO respectively. They evaluate their proposed loss against GRPO and Dr. GRPO losses on the Qwen2.5-Math-7B model. They further conduct ablation studies on Qwen2.5-Math-1.5B model.

**Questions:**

1. From my understanding, $Var(\log(\pi_{\theta}))$ is not related to rewards at all. Then how can one claim in Lines 259-261, that this expression is better than entropy regularization in terms ofensuring undesirable responses receive zero probability, while favorable responses retain comparable probabilities? Are there any experimental validation to this claim?

2. Were more experiments conducted comparing GRPO to GVPO apart from Qwen series of models? This can be crucial to understand the efficacy of the proposed loss. I am willing to change my score based on such evaluation.

**Ethical Concerns:**

["NO or VERY MINOR ethics concerns only"]

**Final Justification:**

New experimental evidence has been provided to further justify the method's efficacy. However, the concerns regarding theoretical novelty and the utility of the variance term in practice still persist. As such, I cautiously increase my score to 4.

**Limitations:**

Yes, but not in the main text.

**Paper Formatting Concerns:**

Only concern is that limitations are not included in the main paper. However, this may be acceptable.

**Quality:**

2

**Strengths And Weaknesses:**

Strengths:
1. A novel loss formulation that incorporates the implicit reward expression from the optimal solution to the KL-constrained reward maximization problem into the GRPO loss framework
2. Clear analysis of the loss and its solution, along with a proper discussion comparing it to DPO and GRPO losses

Weaknesses:
1. Limited Experimental Evaluation (experiments run only on Qwen2.5-Math-7B model)
2. Limited theoretical novelty. Note that the unified loss gradient framework in terms of weights has already been introduced in [1]. The primary novelty in this work seems to be novel weight assignment in terms of implicit and actual rewards. It straightforwardly follows from the definition of implicit rewards that the optimal solution to such a loss is also the optimal solution to the KL-constrained reward maximization problem.



[1] Gao, B., Song, F., Miao, Y., Cai, Z., Yang, Z., Chen, L., ... & Chang, B. (2024). Towards a unified view of preference learning for large language models: A survey. arXiv preprint arXiv:2409.02795.

---

> ### Author Rebuttal · Authors · 2025-07-31
>
> We appreciate your recognition of the novelty in our loss formulation and the clarity of our analysis.
>
> Building on this foundation, we respectfully address the following points of concern to further clarify our contributions.
>
>
> **1. Limited Experimental Evaluation (experiments run only on Qwen2.5-Math-7B model)**
>
> We have included experiments on an additional base model as presented below.
>
> Furthermore, we have expanded our evaluation in the rebuttals by incorporating more benchmarks, conducting ablations on the regularization terms, performing robustness check across seeds, and evaluating an alternative task of summarization.
>
> **2. Limited theoretical novelty.**
>
> **It straightforwardly follows from the definition of implicit rewards that the optimal solution to such a loss is also the optimal solution to the KL-constrained reward maximization problem.**
>
> While the necessity of the theorem ($R_\theta = R$ is an optimal solution) follows directly, establishing its sufficiency (the only optimal solution is $R_\theta = R$) is significantly more challenging and is addressed in detail in Appendix B.1.
>
> The sufficiency result is novel—for instance, DPO establishes only necessity, not sufficiency, and admits multiple optimal solutions, some of which does not correspond to the optimal solution of the KL-constrained reward maximization objective (see Lines 231–238).
>
>
> **$Var(log(\pi_\theta))$ is not related to rewards at all. Why is this expression better than entropy regularization in terms of ensuring undesirable responses receive zero probability, while favorable responses retain comparable probabilities? Are there any experimental validation to this claim?**
>
> We justify this expression both theoretically and empirically.
>
> First, we clarify that $Var(\log(\pi_\theta))$ is intended to ensure some responses receive zero probability, while other responses retain comparable probabilities. As just introduced in the paper, the term $R - \overline{R}$ promotes advantage maximization; therefore, $Var(\log(\pi_\theta))$ serves to ensure that undesirable responses receive zero probability, while favorable responses maintain similar probabilities. We will revise the manuscript to make this point clearer.
>
> To illustrate the benefit of $Var(log(\pi_\theta))$, consider a toy example involving generation over the tokens $a, b, c, d, e$, where $\pi_\theta(a) = \pi_\theta(b) = 0$ and $\pi_\theta(c) = \pi_\theta(d) = \pi_\theta(e) = 1/3$. In this case, $Var(\log(\pi_\theta)) = 0$, indicating minimum variance and thus no penalty on this distribution. In contrast, entropy regularization only reaches its minimum either when the distribution is uniform or when it is one-hot, depending on whether one encourages higher or lower entropy.
>
> We further validate this claim through ablation studies.
>
> | Algorithm           | AIME2024 | AMC    | MATH500 | Minerva | Olympiadbench |
> |--------------------|----------|--------|---------|---------|----------------|
> | GVPO          | 20.72 | 62.65 | 83.80 | 45.95 | 46.96 |
> | GVPO $w/o$ Var          | 0.00 | 0.00 | 0.00 | 0.00 | 0.00 |
> | GVPO $w/o$ Var $w/$ Entropy          | 0.00 | 0.00 | 0.00 | 0.00 | 0.00 |
> | GVPO $w/o$ Var (LR=1e-6)          | 0.00 | 0.00 | 0.00 | 0.00 | 0.00 |
> | GVPO $w/o$ Var $w/$ Entropy (LR=1e-6)          | 3.02 | 39.76 | 33.20 | 23.16 | 8.89 |
>
> In the first set of experiments, we (a) remove the $Var(\log(\pi_\theta))$ term, and (b) replace it with entropy regularization. In both cases, the models fail to converge and produce incoherent outputs, demonstrating that $Var(\log(\pi_\theta))$ plays a crucial role in stabilizing training by discouraging extreme probability assignments.
>
> In the second set, we lower the learning rate to $1\text{e}{-6}$ and repeat the experiments. We observe that removing the term still leads to divergence, whereas replacing it with entropy regularization shows slightly improved but still unsatisfactory performance. These results suggest that while entropy regularization serves a similar purpose, it does not fully substitute $Var(\log(\pi_\theta))$. Moreover, the poor performance may stem from entropy regularization being either too weak—failing to suppress extreme values—or too strong—hindering convergence. This highlights a key drawback of entropy regularization: it requires careful coefficient tuning. In contrast, the coefficient for $Var(\log(\pi_\theta))$ is determined analytically via the optimal solution theorem, thereby avoiding this sensitivity and improving robustness.
>
> **Were more experiments conducted comparing GRPO to GVPO apart from Qwen series of models?**
>
> Yes. We conducted additional experiments using Llama-3.1-8B-Instruct as a new base model. The results consistently show that GVPO outperforms GRPO under this new setting, demonstrating its effectiveness again.
>
> | Algorithm           | AIME2024 | AMC    | MATH500 | Minerva | Olympiadbench |
> |--------------------|----------|--------|---------|---------|----------------|
> | Llama-3.1-8B-Instruct       |  5.00   | 19.27  | 50.40   | 22.42   |  17.04         |
> | +GRPO              | 8.54    | 19.27  | 52.60   | 25.37   | 16.15          |
> | **+GVPO**          | **11.56**| **20.48** | **56.60** | **29.41** | **20.59** |

---

> > ### Comment · Reviewer_KXfz · 2025-08-01
> >
> > Thank you for the additional experiments. It does strengthen the paper and the proposed method.
> >
> > Thank you for clarifying the relevance of $Var(log(\pi_\theta))$ term. I have a better understanding now. The ablation study is also helpful to show the importance of the term $Var(log(\pi_\theta))$.
> >
> > Although I would note that, for the new loss that removes the variance term (or replaces it with an entropy term), the unique minimizer is not the optimal solution to the KL-constrained reward maximization problem. Hence, it is expected that the training doesn't converge and leads to incoherent outputs. The authors' claim in the paper is that $Var(log(\pi_\theta))$ ensures "undesirable responses receive zero probability, while favorable responses retain comparable probabilities". Even though, I understand this from their toy example, I do not see any experimental validation for this claim.
> >
> > Regarding theoretical novelty, I still stand by the opinion that it is quite straightforward after going through the proof of sufficiency as well. It is a squared loss, so it is obvious that there exists a unique minimizer.
> >
> > Considering the new experiments, I will increase the score to 4.

---

> > > ### Author Response · Authors · 2025-08-09
> > >
> > > We sincerely thank you again for your valuable feedback and support, which have greatly helped us improve our work.

---

### Official Review · Reviewer_eBjY · 2025-07-02

**Clarity:** 3
**Significance:** 3
**Originality:** 4
**Rating:** 4
**Confidence:** 4

**Summary:**

This article introduces Group Variance Policy Optimization (GVPO), a new approach for post-training LLMs that ensures reliable and robust model alignment with specific tasks and human preferences. GVPO directly incorporates the analytical solution to KL-constrained reward maximization into its gradient weights, offering two key advantages: a unique optimal solution aligned with the KL-constrained reward maximization objective and support for flexible sampling distributions that avoid limitations of on-policy and importance sampling methods.

**Questions:**

See Weaknesses.

For instance,

1. **Separation of Reward Modeling and Policy Optimization**
Does GVPO require a separate training phase for the reward model before applying the algorithm?
How does the performance of GVPO depend on the quality of the separately trained reward model?

2. **Insufficient Experimentation:**
How can the experimental section be expanded to provide more thorough empirical validation of the proposed method ?

**Ethical Concerns:**

["NO or VERY MINOR ethics concerns only"]

**Final Justification:**

My concerns have been addressed, and I will raise my score.

**Limitations:**

See Weaknesses

**Quality:**

3

**Strengths And Weaknesses:**

**Strengths**:

1. GVPO builds on the insights from Group Relative Policy Optimization (GRPO) but addresses its training instability issues. By designing a weighting scheme where the gradient weight of a response is the difference between the central distance of implicit rewards (derived from the current and reference policies) and actual rewards, GVPO ensures that the sum of weights within a prompt group equals zero. This design eliminates the need for explicit estimation of the partition function.

2. Theoretical guarantees of GVPO are rigorously established. It is proven that GVPO’s loss function is aligned with the original constrained optimization problem, ensuring convergence to the globally optimal policy.

3. A Few experimental results demonstrate GVPO’s effectiveness.

**Weaknesses**:

1. **Separation of Reward Modeling and Policy Optimization**:
    GVPO appears to reintroduce the separation between reward modeling and policy optimization. Unlike DPO, which integrates reward model training and policy optimization into a single process by exploiting their analytical relationship, GVPO seems to require a predefined reward model  $R$  (as indicated in Equation (9), not $R_{\theta} = \frac{\log(\pi_{\theta})}{\log(\pi_{\theta^{'}})}$). This raises questions about whether GVPO necessitates a separate training phase for the reward model before applying GVPO. If so, this could be seen as a limitation, as the performance of GVPO might become heavily dependent on the quality of the separately trained reward model, potentially undermining its overall effectiveness.

 2. **Redundant Presentation of Equations**:
    The presentation of equations in the paper could be streamlined for better clarity. For instance, the two equations introduced between lines 142--152 appear to primarily serve the purpose of supporting the equation presented between lines 159--160 that illustrates that GVPO is equivalent to optimizing a mean squared error (MSE) loss that measures the discrepancy between implicit and actual reward central distances. To enhance the flow, the authors could first introduce the symbol $R_{\theta} = \frac{\log(\pi_{\theta})}{\log(\pi_{\theta^{'}})}$, then present the equation between lines 159--160. Following this, they could explain how the equation between lines 142--152 arises from the condition $\sum w_{i}=0$ for verifying the equation between lines 159--160.

 3. **Redundant Theoretical Proofs in Section 3.3**:
    The theoretical proofs in Section 3.3 feel repetitive, as they focus extensively on retelling the result that GVPO is equivalent to optimizing an MSE loss measuring the discrepancy between implicit and actual reward central distances.


  4. **Misleading Name ("Variance")**:
    The choice of the name "Variance" for the proposed method might be misleading. The new weight introduced in GVPO is actually a difference of two deviations from the the mean rather than a pure variance. This could cause confusion regarding the method's core mechanism and its relationship to statistical variance.

5. **Insufficient Experimentation**:
    Since this paper focuses on proposing a new post-training algorithm rather than being a theoretical reinforcement learning(RL) paper, the experimentation section feels somewhat insufficient. The proposed method lacks thorough empirical validation across diverse LLMs, which would be crucial for demonstrating its effectiveness and generalizability in practical applications.

---

> ### Author Rebuttal · Authors · 2025-07-31
>
> We appreciate your recognition of GVPO’s advancements over GRPO. We also thank you for acknowledging the rigorous theoretical guarantees of our paper.
>
> Below, we address your concerns and provide clarifications to further strengthen our contribution.
>
>
> **1. Separation of Reward Modeling and Policy Optimization**
>
> **Does GVPO require a separate training phase for the reward model before applying the algorithm?**
>
> No, GVPO does not require a separate training phase. GVPO supports flexible sampling distributions, including those derived from human ratings or preferences, similar to DPO, and thus does not necessitate training a separate reward model (Lines 278–283).
>
> In our paper, the reward model can take various forms: it may explicitly consist of human evaluation ratings, a trainable function that implicitly captures human preferences, or a predefined rule—such as correctness or accuracy (Lines 94–95).
>
> **How does the performance of GVPO depend on the quality of the separately trained reward model?**
>
> LLM post-training scenarios generally fall into two categories: reinforcement learning with verifiable rewards (RLVR) and reinforcement learning from human feedback (RLHF).
>
> In RLVR settings, there is no need for a trained reward model. In RLHF, if GVPO uses human-labeled data directly—similar to DPO—it likewise avoids reliance on a reward model's quality. However, when GVPO is trained using a learned reward model as a proxy, its performance, like that of other RL algorithms, will depend on how accurately the proxy reflects true rewards. This proxy reward issue [1] is a well-known challenge in reinforcement learning. Consequently, it is common practice to evaluate post-training algorithms in the RLVR setting.
>
> [1] Skalse, Joar, et al. "Defining and characterizing reward gaming." Advances in Neural Information Processing Systems 35 (2022): 9460–9471.
>
> **2. Redundant Presentation of Equations**
>
> We appreciate your detailed suggestion. We will revise the manuscript to first introduce $R_\theta$, followed by the MSE equations. We will then explain how these equations naturally arise from the condition $\sum w_i=0$.
>
> **3. Redundant Theoretical Proofs in Section 3.3**
>
> Thank you for this helpful suggestion. We will move less essential proofs (such as Lemmas 3.4 and 3.5) to the appendix to allocate more space for experimental results.
>
> **4. Misleading Name ("Variance")**
>
> We apologize for any confusion caused by the use of “Variance” in GVPO’s name. Our intention was to create a counterpart to GRPO’s "Relative" terminology. Specifically, while GRPO’s weight includes a term $R - \overline{R}$, GVPO’s MSE loss $[(R_\theta - \overline{R_\theta}) - (R - \overline{R})]^2$ can be rewritten in a variance-like form: $[(R_\theta - R) - \overline{(R_\theta - R)}]^2$. We will clarify this interpretation in the revised manuscript.
>
> **5. Insufficient Experimentation. How can the experimental section be expanded to provide more thorough empirical validation of the proposed method ?**
>
> We have strengthened our empirical evaluation in the rebuttals by
> + adding experiments on an additional base model (in the Response to Reviewer KXfz)
> + incorporating a broader range of benchmarks (in the Response to Reviewer X8wt)
> + conducting ablation studies on the regularization terms (in the Response to Reviewer u3Wn)
> + performing robustness check across seeds (in the Response to Reviewer u3Wn)
> + including results on an alternative summarization task (in the Response to Reviewer X8wt).

---

> > ### Comment · Reviewer_eBjY · 2025-08-05
> >
> > Thank you for the response. My concerns have been addressed, and I will raise my score.

---

> > > ### Author Response · Authors · 2025-08-09
> > >
> > > We sincerely thank you again for your valuable feedback and support, which have greatly helped us improve our work.

---

### Official Review · Reviewer_u3Wn · 2025-07-02

**Clarity:** 2
**Significance:** 2
**Originality:** 3
**Rating:** 3
**Confidence:** 2

**Summary:**

This paper develops a method for mitigating instability in GRPO, which it claims arises from exploding gradients that are a consequence of minimizing log likelihood, called GVPO. GVPO is a weighted policy gradient objective, where weights sum to zero and are derived from the closed form of the KL-regularized policy objective. There are a number of theoretical results showing that GVPO recovers the unique optimal policy, which rely on a reduction from GVPO to squared-loss regression. Experiments are provided in a number of mathematical reasoning datasets.

**Questions:**

Why does incorporating the log-policy ratio terms into GVPO increase stability?

**Ethical Concerns:**

["NO or VERY MINOR ethics concerns only"]

**Final Justification:**

I have considered the authors' rebuttal and other reviewers' comments; further justification is provided in my response to the rebuttal.

**Limitations:**

See comments above

**Quality:**

2

**Strengths And Weaknesses:**

Overall, I think that the proposed method is interesting, but requires more in-depth empirical investigation. The clarity of the writing, motivation for the paper, and theoretical results are somewhat weak in my view, and as a result I would have really liked to see ablations targeting the purported variance-reduction benefits of GVPO.

**Motivation**

To begin with, I found the motivation for the paper to be underwhelming and largely unsupported. The motivation for the method is that GRPO is unstable because "minimizing the log-likelihood of those responses can lead to exploding gradients due to the convexity of the logarithmic function" (L107-108), but there is no evidence within the paper that this is true, nor citations for this claim.

In addition, the motivation for the method is to reduce policy learning to reward learning (L148, 151-152), but why is this better than optimizing the policy? And what is the relationship with GVPO (Eq. 9) itself?

**Method and theoretical results**

There are interesting manipulations and interpretations of the gradient objective: using policy gradient weights that sum to 0, the partition function no longer needs to be estimate, which allows the objective to be converted to squared loss regression. This seems possibly straightforward given the form of Eq. 9 and the definition of $R_\theta = \log \frac{\pi_\theta}{\pi_{\theta'}}$, and the theoretical results are somewhat simple and repetitive (in my opinion, not generating much more insight) beyond the sq-loss interpretation.

As a result, I wonder if the paper might be made stronger if Sections 3.1 and 3.3 were shortened in favor of more insightful experiments and ablations, discussed below.

**Experiments**

I appreciate that the authors included both theoretical and empirical results, and considered a few datasets. However, it appears that no standard deviations or averages over seeds are reported, which I find discomfiting given the variance of policy gradient methods for training language models (e.g., GVPO, by the paper's own admission). While I understand that better performance can reflect increased stability, I think that the experimental results would be more interesting if they explicitly highlighted the intended benefits of GVPO.

The objective resembles the GRPO objective, except it doesn't divide by the standard deviation, and has an additional pair of log policy-ratio terms that can be viewed as implicit rewards. But it has been observed empirically in DPO that such policy-parameterized rewards can be suboptimal or less stable compared to reward models (Swamy et al. 2025; Lin et al. 2024). As such I am curious about the effect of the policy ratio terms, I would think that there could be interesting ablations to conduct on their role in stability, as well as in probing the predictions in Section 3.5. It's not really clear to me how much benefit comes from removing the standard deviation normalization in GRPO (as discussed in L245-249) versus adding the log policy ratios.

**References**

Swamy, Gokul, Sanjiban Choudhury, Wen Sun, Zhiwei Steven Wu, and J. Andrew Bagnell. "All roads lead to likelihood: The value of reinforcement learning in fine-tuning." _arXiv preprint arXiv:2503.01067_ (2025).

Lin, Yong, Skyler Seto, Maartje Ter Hoeve, Katherine Metcalf, Barry-John Theobald, Xuan Wang, Yizhe Zhang, Chen Huang, and Tong Zhang. "On the limited generalization capability of the implicit reward model induced by direct preference optimization." _arXiv preprint arXiv:2409.03650_ (2024).

---

> ### Author Rebuttal · Authors · 2025-07-31
>
> We appreciate your feedback and the recognition of the potential interest in our method. However, we believe there may be a misunderstanding regarding the motivation and contributions of the paper.
>
> We clarify this further below and address your questions and concerns.
>
>
> **Overall, I think that the proposed method is interesting, but requires more in-depth empirical investigation. ... I would have really liked to see ablations targeting the purported variance-reduction benefits of GVPO.**
>
> We fully understand your concerns. However, we believe there may have been a misunderstanding. This paper does not aim to reduce variance, nor have we claimed that it can. Our primary objective is to propose a better post-training algorithm compared to GRPO. To that end, we have thoroughly compared GVPO and GRPO to highlight the benefits of our approach, and other reviewers have acknowledged this aspect positively.
>
> Upon reflection, we recognize that the term "Variance" in GVPO’s name may have led to confusion. Our goal was to create a counterpart to GRPO's "Relative" term. Specifically, while GRPO's weight contains $R-\overline{R}$ (a relative term), GVPO’s MSE loss $[(R_\theta-\overline{R_\theta})-(R-\overline{R})]^2$ can also be expressed as a variance-like term, $[(R_\theta-R)-\overline{(R_\theta-R)}]^2$. We will clarify this distinction in the revised draft.
>
> **Motivation**
>
> **The motivation for the method is that GRPO is unstable ... but there is no evidence within the paper that this is true, nor citations for this claim.**
>
> Still, we believe there may have been a misunderstanding. The motivation of this paper is not: because GRPO is unstable, so we improve its stability. But instead is: GRPO are reported not effective in practice (and many people attributes to its instability), so we propose a more effective one. In other words, this work focuses on better performance rather than low variance. Framing our motivation as “variance reduction” risks underestimating the true contribution of our work.
>
>
> To clarify, we will add relevant references and provide a more clear explanation in Lines 106-111:
>
> \begin{equation}
> \mathcal{L}(\theta) = -\sum_{(x,\{y_i\}) \in \mathcal{D}} \sum_{i=1}^k \frac{R_i-\overline{R}}{\sigma(R)}\log \pi_\theta(y_i|x)
> \end{equation}
>
> Rewards below average lead to negative weights, penalizing their likelihoods. However, minimizing likelihood can result in unstable training [1]. Additionally, off-policy training of GRPO, which involves importance sampling with the weight $\frac{\pi_\theta}{\pi_{\theta_{old}}}$, becomes unstable when $\pi_\theta$ significantly deviates from $\pi_{\theta_{\text{ref}}}$, potentially causing gradient explosions. To mitigate these issues, GRPO uses gradient clipping and a KL-divergence constraint between the updated and initial policies:
> \begin{equation}
> \mathcal{L}(\theta) = -\sum_{(x,\{y_i\}) \in \mathcal{D}} \sum_{i=1}^k Clip(\frac{\pi_\theta(y_i|x)}{\pi_\text{old}(y_i|x)})\frac{R_i-\overline{R}}{\sigma(R)}\log \pi_\theta(y_i|x)- \beta KL(\pi_\theta||\pi_\text{ref})
> \end{equation}
> Nevertheless, empirical results [2,3] show that GRPO still exhibits training instability, which undermines its performance.
>
>
> [1] Abdolmaleki, Abbas, et al. "Learning from negative feedback, or positive feedback or both." ICLR. 2025.
>
> [2] Yu Q, Zhang Z, Zhu R, et al. Dapo: An open-source llm reinforcement learning system at scale[J]. arXiv preprint arXiv:2503.14476, 2025.
>
> [3] Hu, Jian, Jason Klein Liu, and Wei Shen. "Reinforce++: An efficient rlhf algorithm with robustness to both prompt and reward models." arXiv preprint arXiv:2501.03262 (2025).
>
> **Why is this (reduce policy learning to reward learning) better than optimizing the policy? And what is the relationship with GVPO (Eq. 9) itself?**
>
> Reducing policy learning to "reward learning" is also a key advantage of DPO: it does not replace policy optimization but rather provides a more smart pathway to it. This is because this method simplifies the joint optimization over policies and rewards to a simpler problem focused solely on rewards. This makes the problem more tractable, as it only requires aligning implicit rewards $R_\theta(x, y)$ with the true reward function $R(x, y)$ (Line 215-217). For instance, in joint optimization, the decision of whether to add standard deviation normalization is ambiguous. In contrast, "reward learning" only necessitates designing a loss function that trains $R_\theta$ to align with $R$, on which the question of whether adding standard deviation will depend. GVPO reduces policy learning to "reward learning" through an MSE loss.
>
>
> **Method and theoretical results**
>
> **This seems possibly straightforward given the form of Eq. 9 and the definition of $R_\theta=\frac{\log \pi_\theta}{\log \pi_{\theta'}}$, and the theoretical results are somewhat simple**
>
> While the necessity of the theorem ($R_\theta = R$ is an optimal solution) follows directly, establishing its sufficiency (the only optimal solution is $R_\theta = R$) is significantly more challenging and is addressed in detail in Appendix B.1.
>
> The sufficiency result is novel—for instance, DPO establishes only necessity, not sufficiency, and admits multiple optimal solutions, some of which does not correspond to the optimal solution of the KL-constrained reward maximization objective (see Lines 231–238).
>
> **I wonder if the paper might be made stronger if Sections 3.1 and 3.3 were shortened in favor of more insightful experiments and ablations**
>
> We agree with your suggestion and will prioritize experiments over less important theoretical discussions.
>
> **Experiments**
>
> **It appears that no standard deviations or averages over seeds are reported, ..., I think that the experimental results would be more interesting if they explicitly highlighted the intended benefits of GVPO.**
>
> While the objective of this work is not to reduce variance, we have nonetheless conducted additional experiments using 10 random seeds for both methods on the 1.5B model with $k=4$ as a robustness check:
>
> |Algorithm|AIME2024|AMC|MATH500|Minerva|Olympiadbench|
> |-|-|-|-|-|-|
> | GRPO| 13.59$\pm$1.11 | 44.34$\pm$2.19 | 76.30$\pm$1.35 | 35.40$\pm$1.04 | 38.65$\pm$0.74 |
> | GVPO| 15.56$\pm$1.05 | 45.78$\pm$2.05 | 77.36$\pm$0.98 | 36.03$\pm$1.15 | 39.58$\pm$1.08 |
>
> The results indicate that GVPO consistently outperforms GRPO in terms of performance, while maintaining comparable standard deviations.
>
> **The objective resembles the GRPO objective, except it doesn't divide by the standard deviation, and has an additional pair of log policy-ratio terms that can be viewed as implicit rewards.**
>
> We would like to respectfully point out that your comparison may overlook a critical component of GRPO—the clipped importance sampling weight, $Clip(\frac{\pi_\theta(y_i|x)}{\pi_\text{old}(y_i|x)})$. This clipping mechanism is essential in GRPO to prevent gradient explosion due to large importance weights. While it effectively stabilizes training, it introduces bias into the gradient estimates. In contrast, GVPO does not rely on importance sampling from the outset, thereby avoiding these issues entirely (see Lines 270–285). This represents a significant advantage over GRPO that may not have been fully considered in your comparison.
>
> **But it has been observed empirically in DPO that such policy-parameterized rewards can be suboptimal or less stable compared to reward models (Swamy et al. 2025; Lin et al. 2024).**
>
> Thank you for providing the relevant literature. Our interpretation is as follows: Although DPO employs log-ratios as implicit rewards—thereby functioning similarly to reward models—it is, at its core, a policy that generates responses. It is therefore reasonable to expect that it may be suboptimal for the specific task of reward estimation, especially when compared to reward models explicitly trained to output scalar values.
>
> **As such I am curious about the effect of the policy ratio terms, I would think that there could be interesting ablations to conduct on their role in stability, as well as in probing the predictions in Section 3.5.**
>
> We conduct comprehensive ablation studies on the regularization terms.
>
> |Algorithm|AIME2024|AMC|MATH500|Minerva|Olympiadbench|
> |-|-|-|-|-|-|
> | GVPO| 20.72 | 62.65 | 83.80 | 45.95 | 46.96 |
> | GVPO $w/o$ Var| 0.00 | 0.00 | 0.00 | 0.00 | 0.00 |
> | GVPO $w/o$ Cov| 0.00 | 0.00 | 0.00 | 0.00 | 0.00 |
> | GVPO $w/o$ Var $w/o$ Cov| 7.19 | 39.76 | 73.00 | 34.93 | 35.70 |
>
> First, we remove $Cov(\log\pi_\theta, \log\pi_{\theta^\prime})$ and $Var(\log\pi_\theta)$ individually. In both cases, the models fail to converge and produce incoherent outputs, indicating that these terms are essential for stabilizing training. Next, we remove both regularization terms simultaneously, i.e., removing the ratio terms and reducing the objective to $R - \overline{R}$. While this model initially converges, it begins to diverge after approximately 10% of the training steps, further confirming that the regularization terms are critical components of GVPO.
>
> **Questions**
>
> **Why does incorporating the log-policy ratio terms into GVPO increase stability?**
>
> From the perspective of regularization, the log-policy ratio terms are equivalent to the sum of $Var$ and $Cov$ after taking gradients. These two components help discourage extreme probability assignments and prevent the current policy from deviating excessively from a reference policy. Both effects contribute to improved stability.

---

> > ### Comment · Reviewer_u3Wn · 2025-08-09
> > **Thanks**
> >
> > Thank you for the detailed response. I have slightly increased my score given that it seems the methodology may be useful and of interest to the wider community.
> >
> > While I understand that the goal is to develop a better method than GRPO, I find the motivation (exploding gradients, instability, which is what I meant by variance) and any insight into why the proposed method might lead to improvements in this regard to still be pretty unclear.

---

> ### Author Response · Authors · 2025-08-09
>
> Thank you again for your comments. We have carefully considered them and addressed the concerns in our responses above.

---

### Official Review · Reviewer_riT8 · 2025-07-02

**Clarity:** 4
**Significance:** 1
**Originality:** 3
**Rating:** 5
**Confidence:** 3

**Summary:**

This paper presents a new method for policy optimization for LLM post-training, referred to as Group Variance Policy Optimization (GVPO).
GVPO puts together various ideas from policy optimization and preference learning in a new way.  In particular, via a clever trick that centers the weights of gradients (where the weights are differences in log-likelihood ratios), a partition function can be made to vanish in the gradient.  This yields not only an efficient method, but an interpretable one.

**Questions:**

It would be nice to understand whether there is a more general principle at work here.  As currently presented, the idea (center the weights) comes across as a (nice) trick.  Are there analogous methods in the operations research literature?  Or the optimization literature?

In particular, the objective function studied here is of the form of mirror descent, which may provide a path towards a deeper understanding and even obtaining theoretical convergence rates.

**Ethical Concerns:**

["NO or VERY MINOR ethics concerns only"]

**Final Justification:**

I view this as a valuable contribution to the literature on LLM post-training with a useful and interesting algorithmic innovation of weight-centering.

**Limitations:**

Yes

**Quality:**

4

**Strengths And Weaknesses:**

Strengths:  The authors do an excellent job at situating their method in the landscape of existing methods, and pointing out its relative advantages, which include a lack of a need for importance sampling and on-policy gradients, and therefore a lack of need for various ad hoc adjustments including clipping.  The authors also do an excellent job at interpreting the gradient of their method, in three different ways, all insightful.  The experiments also seem compelling.

Weaknesses:  None that I can see.

---

> ### Author Rebuttal · Authors · 2025-07-31
>
> Thank you for engaging with our work and highlighting the novel aspects we take pride in. We appreciate your recognition of the clarity with which we position our method in relation to existing approaches, as well as your appreciation of the three complementary interpretations of the gradient. We're also pleased that you found the experimental results compelling.
>
> **It would be nice to understand whether there is a more general principle at work here. ... In particular, the objective function studied here is of the form of mirror descent, which may provide a path towards a deeper understanding and even obtaining theoretical convergence rates.**
>
> Thank you for highlighting this promising direction for future research. Based on our current understanding, we are not aware of a unifying principle in OR or optimization literature that directly explains this method. Nonetheless, we share your interest in theoretical convergence rates and find questions—"Which sampling distributions exhibit faster convergence? What characteristics do they share?"—both insightful and significant. We will keep your suggestion regarding "mirror descent" in mind and plan to explore this perspective further in future work.

---

> ### Comment · Reviewer_riT8 · 2025-08-04
>
> Having read the other reviews and the rebuttal I continue to view this as a valuable contribution to the literature on LLM post-traiing and I retain my favorable score.

---

> > ### Author Response · Authors · 2025-08-09
> >
> > We sincerely thank you again for your valuable feedback and support, which have greatly helped us improve our work.

---

### Official Review · Reviewer_X8wt · 2025-07-03

**Clarity:** 4
**Significance:** 3
**Originality:** 3
**Rating:** 5
**Confidence:** 4

**Summary:**

This paper presents Group Variance Policy Optimization (GVPO), a method for KL-constrained reward maximization. To derive GVPO, the paper identifies that when the sum of assigned weights to all responses within a prompt equals zero, then the partition function becomes invariant across the compared responses. The paper presents a clean interpretation as well as connections with existing literature on policy optimization methods. The paper also presents empirical results highlighting the usefulness of the proposed policy search method on math reasoning tasks.

**Questions:**

While the paper presents clear benchmarking for math reasoning tasks, it is worth asking the broader question -- what is the role of policy optimization in general natural language generation tasks? Does better policy optimization lead to better alignment? This question is more nuanced since there are other factors at play about quality of reward models, how they align with human preferences, reward hacking etc. I think the paper benefits a discussion surrounding these issues, and additional experimentation to highlight the nuanced nature of the discussion surrounding development of better policy search methods. Towards this, I'd like to see (a) additional results on natural language generation tasks such as summarization, question answering etc., (b) whether a better policy search method naturally translates to better evals on automated/human eval on alignment quality. My sense is these results will be illuminating and may not fully line up in terms of improvement in metrics, yet will be beneficial to guide subsequent research in this topic.

**Ethical Concerns:**

["NO or VERY MINOR ethics concerns only"]

**Final Justification:**

The authors agree to expanding their results to handle alignment benchmarks and engaged with addressing comments about insufficient benchmarking. I have updated my score based on these discussions.

**Limitations:**

yes

**Paper Formatting Concerns:**

None.

**Quality:**

3

**Strengths And Weaknesses:**

Strengths:
- Important problem, well motivated and well written paper
- Clearly well situated with important (and most well known) related works such as GRPO, DPO etc.
- clear derivation and strong empirical results

Weaknesses:
- Insufficient empirical benchmarking

---

> ### Author Rebuttal · Authors · 2025-07-31
>
> We appreciate your recognition of the importance and clarity of our work, as well as the relevance of our positioning within the literature. We are particularly encouraged by the acknowledgment of our theoretical derivations and empirical results.
>
> Below, we address your concerns and provide clarifications to further strengthen our contribution.
>
> **Insufficient empirical benchmarking**
>
> We have added three more recently popular benchmarks, including DAPO [1], Remax [2], and Reinforce++ [3].
>
> | Algorithm           | AIME2024 | AMC    | MATH500 | Minerva | Olympiadbench |
> |--------------------|----------|--------|---------|---------|----------------|
> | Qwen2.5-Math-7B     | 14.68    | 38.55  | 64.00   | 27.20   | 30.66          |
> | +GRPO              | 14.79    | 55.42  | 80.00   | 41.17   | 42.07          |
> | +Dr.GRPO           | 16.56    | 48.19  | 81.20   | 44.48   | 43.40          |
> | +DAPO              | 3.23     | 20.48  | 50.40   | 28.68   | 16.59          |
> | +Remax             | 17.19    | 60.24  | 82.00   | 40.44   | 45.19          |
> | +Reinforce++       | 16.67    | 54.22  | 80.40   | 43.01   | 41.78          |
> | **+GVPO**          | **20.72**| **62.65** | **83.80** | **45.95** | **46.96** |
>
> The results demonstrate that GVPO continues to retain its effectiveness.
>
> [1] Yu Q, Zhang Z, Zhu R, et al. Dapo: An open-source llm reinforcement learning system at scale[J]. arXiv preprint arXiv:2503.14476, 2025.
>
> [2] Li, Ziniu, et al. "Remax: A simple, effective, and efficient reinforcement learning method for aligning large language models." arXiv preprint arXiv:2310.10505 (2023).
>
> [3] Hu, Jian, Jason Klein Liu, and Wei Shen. "Reinforce++: An efficient rlhf algorithm with robustness to both prompt and reward models." arXiv preprint arXiv:2501.03262 (2025).
>
> **I think the paper benefits a discussion surrounding these issues, and additional experimentation to highlight the nuanced nature of the discussion surrounding development of better policy search methods. Towards this, I'd like to see (a) additional results on natural language generation tasks such as summarization, question answering etc., (b) whether a better policy search method naturally translates to better evals on automated/human eval on alignment quality.**
>
> Thank you for your valuable suggestions, which help broaden the impact of this work within the alignment research community.
>
> From a theoretical perspective, when an RL algorithm is trained using a proxy reward model instead of the true reward (e.g., actual human preferences), its performance on the true objective depends critically on the fidelity of the proxy [4]. This challenge of proxy rewards is well recognized in reinforcement learning literature [5].
>
> [4] Wang, Binghai, et al. WorldPM: Scaling Human Preference Modeling. arXiv preprint arXiv:2505.10527, 2025.
>
> [5] Skalse, Joar, et al. Defining and characterizing reward gaming. Advances in Neural Information Processing Systems 35 (2022): 9460–9471.
>
> Following your advice, we conducted additional experiments on a summarization task. Specifically, we used the Reddit dataset from [6] and the reward model OpenAssistant/reward-model-deberta-v3-large-v2. First, we fine-tuned Qwen2.5 1.5B using the dataset to obtain an SFT reference model. We then applied both DPO and GVPO for post-training. The training data was sampled from the reference model and scored using the reward model.
>
> For evaluation, we considered:
>
> + The average reward of generated summaries on the test split.
>
> + Win rate against human-written demonstration answers, as scored by the reward model.
>
> + Preference accuracy using human-labeled comparison data: both DPO and GVPO are trained via
> $R_\theta(x, y) = \beta \frac{\log \pi_\theta(y|x)}{\log \pi_{\text{ref}}(y|x)} + \beta \log Z(x)$. Given a preference pair $(y_w, y_l)$ where $y_w$ is preferred, we compute whether
> $\frac{\log \pi_\theta(y_w|x)}{\log \pi_{\text{ref}}(y_w|x)} > \frac{\log \pi_\theta(y_l|x)}{\log \pi_{\text{ref}}(y_l|x)}$.
>
> | Algorithm       | Reward | Win Rate | Accuracy |
> |----------------|--------|----------|----------|
> | SFT            | 2.84   | 41.85%    |   ---    |
> | +DPO           | 4.83  | 68.28%      | 60.43%    |
> | +GVPO          | 5.75  | 79.49%      | 64.93%    |
>
> The results indicate that GVPO outperforms DPO across these metrics, supporting the view that improvements in policy search methods are positively correlated with alignment quality.
>
> [6] Stiennon, Nisan, et al. Learning to summarize with human feedback. Advances in Neural Information Processing Systems 33 (2020): 3008–3021.

---

> > ### Comment · Reviewer_X8wt · 2025-08-05
> > **Response to author rebuttal**
> >
> > Thank you to the authors for their response. I personally think the response goes in the direction of addressing some of the shortcomings I raised. However, it doesn't go all the way towards fixing these issues. Can the authors confirm if:
> > (a) they can include these results (beyond math benchmarks) in the final version? this includes a discussion surrounding the role of policy optimization within the broader alignment problem?
> > (b) can the authors actually run benchmarking where they query some powerful/large model (or human evals) where they ask for the model/human to judge the quality of the generations? I personally dont believe average reward / win rate / accuracy as defined by the authors goes about addressing the concerns with benchmarking better policy optimization for alignment. I think for the alignment problem, the evaluation needs to be conducted in a more rigorous manner, as it relies on situations where we go beyond solving a math problem.
> >
> > If they confirm that they will do this in the final version of this paper, I will increase my score.

---

> > > ### Author Response · Authors · 2025-08-05
> > >
> > > Thank you for following up and acknowledging our efforts to address the initial concerns.
> > >
> > > We confirm that:
> > >
> > > + (a) We will include all the additional results and discussions on the role of policy optimization within the broader alignment problem.
> > >
> > > + (b) We will run benchmarking by a set of powerful LLMs and human evaluations by formally hiring crowdsourcing workers.
> > >
> > > We are committed to improving the paper in the final version accordingly.

---

> > > > ### Comment · Reviewer_X8wt · 2025-08-05
> > > > **Response to author comment**
> > > >
> > > > Thanks, I will increase my score.

---

> > > > > ### Author Response · Authors · 2025-08-09
> > > > >
> > > > > We sincerely thank you again for your valuable feedback and support, which have greatly helped us improve our work.

---

### Note · Authors · 2025-08-12

We thank the reviewers and the AC for their time and constructive feedback. Below, we briefly summarize our paper and the rebuttal process:

**Summary of the Paper:**

This paper proposes GVPO, a method for large language model post-training. GVPO avoids the importance sampling weights that can cause instability in off-policy policy gradient methods such as GRPO, while still guaranteeing a unique optimal solution to the KL-constrained reward maximization objective. Furthermore, GVPO supports flexible sampling distributions, enabling DPO-style training directly with human-labeled data.

**Summary of the Rebuttal:**

+ Reviewers frequently acknowledged GVPO’s originality and quality. In addition, GVPO appears to have a broader impact that extends beyond the LLM reasoning community.

+ The paper was frequently recognized for its clear and well-grounded positioning relative to related methods, including GRPO and DPO.

+ Initial concerns about the insufficient experimental evaluation—arising because the work primarily relies on deduction (theorems and analytical comparisons)—were addressed by incorporating the suggested experiments in the rebuttal. All of the reviewers who raised these concerns subsequently increased their scores, acknowledging the strengthened empirical validation.

We believe the final version of the work addresses all major concerns and provides both a theoretically sound and practically useful advancement for LLM post-training.

---

### Decision · Program_Chairs · 2025-09-17

**Decision:**

Accept (poster)

**Comment:**

Building upon previous work of GRPO, the paper utilizes multiple rollouts per prompt for robust advantage evaluation in RLHF. The paper proposed to use the squared difference in the central distance of implicit rewards and the actual rewards as the minimization objective and proved several nice properties of such optimization objective including: (1) the solution is the minimizer of a KL regularized reward maximization problem; (2) the method is robust to mis-specification of the sampling distribution by avoiding the importance sampling term. The derivation relies on the zero-sum property of the group calibrated rewards to avoid the computation of the partition term for reward-tilted distribution, which is a nice insight and could be of independent interest. The paper also demonstrates empirically that the method outperforms GRPO, and a few other baseline methods (a few added during discussion) on standard benchmarks.

A majority (4 out of 5) of the reviewers like solid theoretical justification of the method and its strong empirical performance against other baselines. During the discussion phase, the authors further addressed reviewers' concerns on the empirical results on the paper, including providing comparison to other baseline algorithms, experiments on non-math tasks, ablation studies on different characteristics of the algorithm, and benchmarks on non-Qwen models. Most of the reviewers liked the rebuttal and mentioned that their concerns were addressed if the authors could include these new experiments in the revised version.

Reviewers also raised the concern that the presentation of the paper could be improved by shortening the redundant equations in the main proof and add more ablations and empirical validations of the algorithm. I agree with the reviewers' suggestion that the presentation of the theoretical part needs some shortening for precise presentation.

After rebuttal, one reviewer raised a concern about the lack of solid theoretical justification on why the method works, which is a fair point. But I believe this point doesn't outweigh the paper's theoretical and empirical contributions.

Overall, I believe the paper is a solid contribution to the field, and hence I will recommend acceptance. The authors should include the new experiments provided in the discussion and revise their paper based on the suggestion provided by the reviewers.